# Phage Genome Diversity in a Biogas-Producing Microbiome Analyzed by Illumina and Nanopore GridION Sequencing

**DOI:** 10.3390/microorganisms10020368

**Published:** 2022-02-04

**Authors:** Katharina Willenbücher, Daniel Wibberg, Liren Huang, Marius Conrady, Patrice Ramm, Julia Gätcke, Tobias Busche, Christian Brandt, Ulrich Szewzyk, Andreas Schlüter, Jimena Barrero Canosa, Irena Maus

**Affiliations:** 1System Microbiology, Department Bioengineering, Leibniz Institute for Agricultural Engineering and Bioeconomy (ATB), Max-Eyth-Allee 100, 14469 Potsdam, Germany; kwillenbuecher@atb-potsdam.de; 2Environmental Microbiology, Faculty of Process Sciences, Institute of Environmental Technology, Technische Universität Berlin, Ernst-Reuter-Platz 1, 10587 Berlin, Germany; ulrich.szewzyk@tu-berlin.de (U.S.); Jimena.barrerocanosa@tu-berlin.de (J.B.C.); 3Center for Biotechnology (CeBiTec), Genome Research of Industrial Microorganisms, Bielefeld University, Universitätsstr. 27, 33615 Bielefeld, Germany; dwibberg@cebitec.uni-bielefeld.de (D.W.); tbusche@cebitec.uni-bielefeld.de (T.B.); aschluet@cebitec.uni-bielefeld.de (A.S.); 4Faculty of Technology, Bielefeld University, Universitätsstr. 25, 33615 Bielefeld, Germany; huanglr@cebitec.uni-bielefeld.de; 5Institute of Agricultural and Urban Ecological Projects, Berlin Humboldt University (IASP), Philippstr. 13, 10115 Berlin, Germany; Marius.conrady@iasp.hu-berlin.de (M.C.); Patrice.ramm@iasp.hu-berlin.de (P.R.); 6Biophysics of Photosynthesis, Institute for Biology, Humboldt-Universität zu Berlin, Philippstrasse 13, 10115 Berlin, Germany; Julia.gaetcke@hu-berlin.de; 7Institute for Infection Medicine and Hospital Hygiene, University Hospital Jena, Kastanienstraße 1, 07747 Jena, Germany; Christian.Brandt@med.uni-jena.de

**Keywords:** virome, phage particle extraction protocol, phage enrichment, virome structure, bacteriophages, fragment recruitment

## Abstract

The microbial biogas network is complex and intertwined, and therefore relatively stable in its overall functionality. However, if key functional groups of microorganisms are affected by biotic or abiotic factors, the entire efficacy may be impaired. Bacteriophages are hypothesized to alter the steering process of the microbial network. In this study, an enriched fraction of virus-like particles was extracted from a mesophilic biogas reactor and sequenced on the Illumina MiSeq and Nanopore GridION sequencing platforms. Metagenome data analysis resulted in identifying 375 metagenome-assembled viral genomes (MAVGs). Two-thirds of the classified sequences were only assigned to the superkingdom *Viruses* and the remaining third to the family *Siphoviridae*, followed by *Myoviridae*, *Podoviridae*, *Tectiviridae*, and *Inoviridae*. The metavirome showed a close relationship to the phage genomes that infect members of the classes *Clostridia* and *Bacilli*. Using publicly available biogas metagenomic data, a fragment recruitment approach showed the widespread distribution of the MAVGs studied in other biogas microbiomes. In particular, phage sequences from mesophilic microbiomes were highly similar to the phage sequences of this study. Accordingly, the virus particle enrichment approach and metavirome sequencing provided additional genome sequence information for novel virome members, thus expanding the current knowledge of viral genetic diversity in biogas reactors.

## 1. Introduction

Bacteriophages or phages, viruses that infect *Bacteria* and *Archaea,* are increasingly being recognized as the most abundant biological entities on earth, with an estimated amount of 10^31^ particles on our planet [1,2]. Moreover, they play a critical role in shaping microbial diversity and composition. For instance, it is estimated that approximately 20% of the oceanic microbial biomass is lysed by phages daily [3,4]. Phage-mediated lysis affects microbial communities by modulating the abundance of the dominant bacterial species (e.g., kill the winner dynamics) and by releasing high contents of dissolved organic matter, thereby altering nutrient fluxes (i.e., the viral shunt) [5]. Moreover, bacteriophages encode “auxiliary metabolic genes” (AMGs) that may provide new genetic traits to their hosts and/or alter bacterial metabolism [6,7,8,9]. Accordingly, they may affect the performance of the host and its surrounding community. Although the role of phages as modulators of microbial populations has been clearly described, most studies were conducted in marine aerobic environments or without considering the availability of oxygen as a variable that may affect the interaction between the virus and its host [10]. Less research has been conducted on phages that infect anaerobic bacteria and host–phage behavior under anoxic conditions [10]. Among the studies on the anaerobic digestion (AD) of biomass, the ecological relevance of phages for corresponding microbiomes has mainly been studied in the human gut and soil environments, and in the termite gut environment [11,12,13,14,15]. Knowledge about the diversity, taxonomy, ecology, and biology of phages in artificial AD systems, such as AD reactors operated for biogas production, is even more limited due to challenges in working with anaerobic microorganisms. In the context of sustainable energy production, biogas has become an alternative to fossil fuels, complementing other renewable energy sources, such as wind and the sun [16]. Therefore, it is an important part of bioeconomic strategies for carbon dioxide (CO_2_)-neutral energy recovery by the biomethanization of renewable raw materials, optimally gaining from the by-products or wastes of agricultural processes or landscape management. Several metagenome-sequencing studies on biogas microbiomes have proven the presence of virus/phage sequences, thus indicating the importance of the virus community in the AD process. Usually, the prevalence of virus/phage sequences varies between 0.1 and 1% of the entire biogas microbiome [17,18].

Numerous studies have examined the composition of biogas microbiomes by applying omics technologies [17,19,20,21]. These include studies that particularly emphasized the hybrid approach applying Illumina sequencing accompanied by long reads derived from the Oxford Nanopore Technologies (ONT) sequencing platform [22]. Using a hybrid approach, Overholt and colleagues recovered nearly fourfold more viral genomes than the Illumina-only approach after the assembly process, indicating the importance of this approach for studies of microbiomes [23]. The obtained data provided high-resolution insights into the bacterial and archaeal community structures of AD systems and their correlations with process parameters. However, the majority of biogas microbiome members are still unknown [24,25,26]. Other studies also demonstrated that the composition of biogas microbiomes was shaped by different parameters, such as fed substrates, temperature, reactor performance, fermentation type [27,28], or even bacteriophages [10]. Therefore, there is increasing interest in assessing the influence of phages on the microbial population dynamics in AD or biogas plants in particular [17,29].

Recent studies on the purification of virus-like particles and their morphological analysis by transmission electron microscopy (TEM) provided insights into virome compositions originating from biogas plants [29,30]. The majority of identified phages showed well-known viral morphologies. Filamentous viruses featuring flexuous filaments similar to *Inovirus*-like (family *Inoviridae*) or Tymovirales (family *Tymoviridae*) and rigid rod-shaped particles, e.g., *Tobamovirus*-like (family *Virgaviridae*), intermixed with tailed and tail-less icosahedral virus-like particles of different size ranges were observed. Head-tailed and tail-less icosahedral phages infecting both *Bacteria* and *Archaea* have previously been described in methanogenic environments [31,32]. Moreover, *Siphoviridae*-like head-tailed viruses have been reported for methanogenic reactors [33,34].

Our knowledge of virus/phage diversity in AD systems has been improved due to advances in viral metagenomics efforts. Likewise, advances in bioinformatics tools exploring metagenome data have enabled a more in-depth analysis of virus diversity. Corresponding studies used different procedures for sample preparation. However, few of them included steps for the concentration of viral particles before DNA extraction and shotgun sequencing to generate a virus-specific (virome) dataset [29,35].

Calusinska and colleagues presented a detailed characterization of dsDNA- and RNA-viruses obtained from biogas plants and observed their huge genetic diversity in nine different fermentation samples [29]. The authors primarily identified head-tailed phages of the order *Caudovirales*, constituting 77% of all virus sequences identified and analyzed in their study. The most abundant phages in the studied sample were taxonomically assigned to the families *Myoviridae*, *Siphoviridae,* and *Podoviridae*, accounting for 13.2%, 51.5%, and 6.9% of all classified dsDNA viruses, respectively. However, thousands of viral sequences were still left that did not share a detectable homology with reference phage genomes deposited in nucleotide sequence databases. Consequently, limited data on biogas viromes impede the understanding of the importance of phages on biogas microbiomes and the entire biogas process. Applying a microarray-based approach, Zhang and colleagues found *Enterobacteria* phages (family *Myoviridae*) that were most abundant in Chinese full-scale wastewater treatment plants [36]. This observation corresponds to previous findings described by Calusinska et al. [29]. Analyses of metaproteome data originating from eleven biogas reactors revealed that about 1.6% of all identified mass spectra referred to predicted phage proteins [17]. These results clearly showed that phages are not only an integral part of biogas microbiomes, but that they are also very active.

The overall aim of this study was to extend the reference repository with new viral genome information in the context of AD microbiomes. Furthermore, the present work also provides an adapted protocol for the enrichment of bacteriophages from biogas microbial communities as preparation for phage–metavirome studies. Based on virome sequence data, phages were taxonomically classified and functionally analyzed in the context of the AD process. Moreover, the genetic relatedness of phage genomes originating from a biogas microbiome was compared with metagenome sequence information originating from different biogas plants deposited in the NCBI database to deduce their genetic repertoire, specifically emphasizing functions of importance regarding the biogas reactor environment.

## 2. Materials and Methods

### 2.1. Reactor Set-Up and Sampling

A continuous stirred tank reactor (CSTR) with a working volume between 16 and 17 L (24 L total volume) was investigated during this study. The biogas reactor was operated under mesophilic conditions, with the temperature maintained at 38 °C by a heated water jacket. Seed sludge obtained from a local biogas plant (Kaim Agrar-Energie, Nauen, Germany) was initially used for the inoculation of the laboratory-scale reactor and was described in a previous study on the semi-continuous anaerobic digestion of whole-crop cereal silage (wheat: rye [1:1], DAH Energie, Oberkrämer, Germany) at an organic loading rate of 3.5 g organic dry mass L^−1^ d^−1^ over a period of 12 months (unpublished data). After a subsequent rest period of about 6 months including stirring under mesophilic conditions, the whole reactor content was removed from the CSTR and particles >5 mm were separated by a food mill. The CSTR was then re-inoculated with this resulting slurry and supplemented with 24 g of microcrystalline cellulose (MCC, Roth, Karlsruhe, Germany) serving as a standardized substrate. After 7.25 d of operation, additional supplementation with 82.5 g sodium carboxymethyl cellulose (CMC, Roth) was carried out. Biogas production was continuously monitored using a volumetric gas counter (promethano GVSM-Siphon Type II, BlueMethano, Berlin, Germany) (Appendix A) and resulted, after substrate additions, in biogas yields of 735 mL g_ODM_^−1^ (MCC) and 92 mL g_ODM_^−1^ (CMC). The methane concentration was measured in the gas sample collected in gas-tight bags (Tesseraux, Bürstadt, Germany) using a gas analyzer (Multitec 540, Sewerin, Gütersloh, Germany). Additional monitoring comprised continuous measurement of temperature and pH (Pt1000I-S & EGA161 L/PG-S, Meinsberg, Waldheim, Germany), impeller torque (RZR 2102 control, Heidolph, Schwabach, Germany), and the weight of the reactor content (PW12C3, HBM, Darmstadt, Germany). Sampling for phage enrichment was conducted on day 16.9 after MCC addition.

### 2.2. Electron Microscopy

To investigate the morphology of the viral particles, transmission electron microscopy (TEM) was performed. The samples were treated according to the protocol of Ackemann with small modifications [37]. Therefore, 2 mL of the enriched (by centrifugation steps referring to protocol 2.4 and filtered with 0.8 µm CLearLine^®^ polypropylene filters) biogas reactor sample mixed with 2 mL of SMG Buffer (50 mM Tris-HCl p.H 7.5, 100 mM NaCl, 8.1 mM MgSO4, and 0.01% (*w*/*v*) gelatin) was centrifuged at 18,000× *g* for 60 min. The pellet was washed twice and resuspended with 1.5 mL of 0.1 M ammonium acetate [37]. The samples were stored at 4 °C until processing. For staining, 50 µL of the solution was pipetted on a 100-mesh copper grid coated with 0.4% (*w*/*v*) formvar in dichloroethane for absorption for 1 min, and negatively stained with one drop of 1% (*w*/*v*) potassium phosphotungstate. The incubation lasted for 10–20 s. The copper grids were analyzed with two microscopes, a Philips BioTwin CM120 (F.E.I. Company, Hillsboro, OR, USA) at approximately 60 kV, and images were taken using the DITABIS Imaging Plate system and the FEI Tecnai G^2^ 20 S-TWIN (F.E.I. Company, Hillsboro, OR, USA) at 200 kV. The TEM pictures were manually improved by using Adobe Illustrator for contrast increasing and sharpening.

### 2.3. Enrichment and Purification of Phage Particles from a Biogas Microbiome

A sample of the biogas reactor content was mixed with a 1:1 volume of SMG Buffer and 2 g of glass beads of the size Φ5 mm. According to the protocol of Calusinska et al., 2018, the sample was then shaken for 30 min at 250 rpm to detach the phage-like particles from the plant fibers [27]. Subsequently, the sample was centrifuged for 1 h at 48,000× *g* (Beckman L-70 Ultracentrifuge) to sediment large-sized material. The supernatant was sequentially filtered with 6 µm, 0.8 µm, 0.45 µm (CLearLine^®^ Polypropylene filters), and 0.22 µm pore size sterile filters (ROTILABO^®^ CME (mixed cellulose esters)). The flow-through was centrifuged at 43,000× *g* and 4 °C for 4 h (Beckman L-70 Ultracentrifuge, Krefeld, Germany). The pellet was resuspended with 1.5 mL of the BE buffer of the Macherey Nagel DNA Extraction kit NucleoSpin Microbial DNA Mini Kit Machery Nagel, Düren, Germany) to generate a pellet with a high number of phages.

### 2.4. Phage DNA Extraction

The DNA extraction was performed by applying the NucleoSpin Microbial DNA Mini Kit for DNA (Machery Nagel, Düren, Germany) with small modifications. The input of the sample was increased to 1.5 mL. Therefore, the first three steps of the protocol (prepare and lyse the sample and adjust the DNA binding conditions) were repeated five times to increase the efficiency. The DNA concentration was determined by fluorometric quantification using a QubitTM (Invitrogen, Life Technologies, Darmstadt, Germany).

### 2.5. MinION Library Preparation and Sequencing

The extracted DNA was subjected to an additional RNase treatment and subsequent clean up using a Genomic DNA Clean & Concentrator (gDCC) kit (Zymo Research Europe GmbH, Freiburg, Germany) according to the manufacturer’s instructions. In total, 5 ng of phage DNA was required to prepare the Oxford Nanopore Technologies (ONT) rapid PCR genomic library using the Rapid PCR Barcoding Kit (SQK-RPB004) (Oxford Nanopore Technologies GmbH, Oxford Science Park, UK). Sequencing was performed on an Oxford Nanopore Technologies GridION Mk1 sequencer at the CeBiTec (Center for Biotechnology), Bielefeld University (Bielefeld, Germany), using an R9.4. flow cell running for 48 h.

### 2.6. Illumina Library Preparation and Sequencing

For short-read sequencing purposes, 100 ng of the total phage DNA was used. The Illumina sequencing library was constructed using the Nextera DNA Flex Library Prep/Illumina DNA Prep Kit (Illumina Inc., Eindhoven, The Netherlands) according to the manufacturer’s instructions. Subsequently, sequencing of the library was performed on the Illumina MiSeq system applying the Illumina MiSeq Reagent Kit v3 following the 2 × 300 nt indexed high output run protocol.

### 2.7. Base Calling, Read Processing, and Assembly

The base calling and read processing of the nanopore data, as well as the virome assembly, were performed as recently described with small modifications [38]. In brief, MinKNOW (v4.2.10) was used to control the sequencing run protocol followed by base calling with bonito (v0.3.0). Virome assembly was performed using canu v2.1.1 [39]. The resulting contigs were then polished with the Illumina data using Pilon [40], which was run for ten iterative cycles. Bwa-MEM 0.7.12 [41] was used for Illumina read mapping for the first five iterations, and bowtie2 v2.3.2 [42] was used for the second set of five iterations. In the next step, the resulting contigs were filtered by different methods and manual curation of the functional annotation. The final metagenome-assembled viral genome (MAVG) data set only contained contigs with genes typical for phages.

### 2.8. Virome Filtering, Annotation, and Genome Analysis

For the initial analysis, the “What the Phage” (WtP) pipeline, an easy–to–use and parallelized multitool approach for phage identification combined with annotation and classification applying the program HostPhinder [43], was used [44]. Based on the results obtained from this pipeline, the data were manually inspected and filtered for phage contigs. The final virome was annotated by means of prokka v1.13.7 with the virus reference dataset [45]. For further manual annotation and genome analysis, phage genome sequences were imported into GenDB 2.0 [46]. The ORFs of these phage genomes of interest were individually blasted again with the BLASTp of the NCBI database. The visualization of phage genomes was performed using the Snapgene Viewer software, and the visualization of graphs was performed applying the R package ggplot2 [47]. Network analysis was performed applying the vConTact 2.0 network by using ProkaryoticViralRefSeq 94 [48,49]. The network visualization was generated using Cytoscape 3.8.2 [50]. The lifestyle of the phage community was predicted by applying the Phage Classification Tool Set (PHACTS), which is using a local database and a so-called Gini coefficient, that measures how important a protein is towards classifying a phage’s lifestyle [51].

### 2.9. Fragment Recruitment

To determine the distribution and abundance of the phage genomes in different biogas communities, fragment recruitments were performed. To this end, the tool SparkHit [52] was used as described previously [53] and genome sequences of each MAVG were employed as a template. Corresponding computations were scaled-up and parallelized by using the de.NBI Cloud environment (https://www.denbi.de/cloud, accessed on 12 September 2021). SparkHit was applied on metagenome FASTQ files from 66 downloaded biogas metagenome datasets, clearly referenced as a biogas plant at the ENA (www.ebi.ac.uk/, accessed on 8 September 2021). Randomly chosen two million reads of each FASTQ file were compared with each single MAVG genome. The alignment identity threshold was set to >50% similarity. The result of the fragment recruitment was visualized using R.

## 3. Results and Discussion

### 3.1. Occurrence of Virus-Like Particles in the Studied Biogas Reactor by Means of Transmission Electron Microscopy

To analyze the presence of virus-like particles being part of the biogas-producing microbial community, a fermentation sample was directly taken from the mesophilic (38 °C) laboratory-scale biogas reactor. The sampled biogas reactor operated under wet fermentation conditions, characterized by high liquid and relatively low total solid percentages (approx. 5% dry solids). The activity of the microbiome was ensured by biogas production from cellulose derivatives prior to sampling for phage enrichment. The digestion of the commonly used reference substrate microcrystalline cellulose (MCC) resulted in biogas production of 735 mL g^−1^ organic dry mass observed at 7.25 days. This is consistent with the range of biogas formation proposed by the VDI guideline for MCC (Appendix A) [54]. Subsequent supplementation of the cellulose derivative carboxymethyl cellulose (CMC) yielded only 92 mL biogas g^−1^ organic dry mass after 9.65 days (Appendix A). In contrast with MCC, the synthetic compound CMC has been described to show virtually any activity in anaerobic digestion [55].

A digestate sample was taken and examined for the occurrence of phage-particles by means of transmission electron microscopy (TEM). The TEM analysis revealed phages with the morphology of *Myoviridae* (phages with long contractile tails) (Figure 1A,E), icosahedral phage-like particles (Figure 1B), *Podoviridae* (short non-contractile tails) (Figure 1C), and *Rudiviridae* (rigid rod-like particles) (Figure 1D). Unfortunately, the TEM analysis of phage-like particles from biogas digestate is often a challenging task because of sample impurities (e.g., undigested plant fibers or corn residues). Therefore, phenotypic phage particle assignment is often only possible to a limited extent. This is in line with the previous study characterizing viral morphologies in AD reactors by means of TEM including wastewater treatment plants, farm digesters, and biogas units treating a mixture of municipal solid waste and agro-food residues [29].

### 3.2. Virome Separation, Enrichment, Sequencing, and Bioinformatic Analysis

To gain deep insights into the taxonomic composition of the phage community obtained from the studied biogas reactor, an extracted fraction enriched for virus-like particles was sequenced on the Illumina MiSeq and Nanopore GridION sequencing platforms. The Illumina data were used to improve the base accuracy, and thus significantly reduce the error rates in the obtained sequences. The Nanopore sequencing run generated 4.04 million reads, yielding a total of 15.47 Gb. The mean read length averaged 4140 bp, with a maximum read length of 8723 bp. The Illumina sequencing run (2 × 300 bp) resulted in 32.3 million reads, yielding 9.7 Gb. After the initial tests, the diversity of the dataset was lower than expected. To avoid oversampling problems in the assembly, the Nanopore dataset was shrunk to 56,000 reads, with 208 Mb. The following canu assembly with the shrunk data set generated 629 contigs with a total size of 5.3 Mb. These contigs were polished by the Illumina data and pilon, followed by annotation using the WtP pipeline and PROKKA. In total, 375 contigs representing the metagenome-assembled viral genomes (MAVGs) could be identified and selected for further analyses, whereas the remaining contigs were filtered out. The final virome had a size of 4.61 Mb and included 7117 predicted genes and 7 tRNAs. The annotation was manually inspected and refined in GenDB 2.0 [46]. The read coverage per contig showed a large difference between 1-fold and 9083-fold, representing the abundance range of viral genomes within the biogas virome. Details of the MAVGs genomes are listed in the Appendix A.

### 3.3. Phage Diversity in the Studied Biogas Plant as Analyzed by Means of Viral Metagenome Data

In addition to the TEM analysis described earlier in this Section 3.1., the sequencing approach and bioinformatics analysis revealed 375 MAVGs representing individual phage genomes, with the 30 most abundant and two largest MAVGs listed in Table 1. In total, 104 of the 375 MAVGs could only be classified to the superkingdom *Viruses*, indicating that one third of the viral sequences originating from the investigated biogas reactor were so far unknown and represented putative novel viral genetic diversity (Appendix A). However, 269 of the 375 MAVGs were assigned to the order *Caudovirales*, with the most abundant families within this order being *Siphoviridae* (133 MAVGs), *Myoviridae* (101 MAVGs), and *Podoviridae* (28 MAVGs), followed by the remaining seven MAGs of the order *Caudovirales*, which were not classifiable to any other known family (Figure 2). In addition, MAVGs 157 and 515 were assigned to the families *Inoviridae* (class *Faserviricetes*) and *Tectiviridae* (class *Tectiliviricetes*), respectively. At the genomic level, *Caudovirales* are an highly diverse group featuring mosaic genomes composed of conserved and variable regions [30]. The predominance of this order in biogas-producing plants was reported previously [36,56,57] and is now confirmed by this study.

Within the family *Siphoviridae,* MAVG 80 was particularly noticeable since it dominated the phage community. Its genome showed a 9083-fold coverage by metagenomic reads, indicating that MAVG 80 is one of the most abundant phage species within the biogas virome analyzed. Members of the family mentioned above represent phages with dsDNA packaged into a capsid connected to a tail [33]. *Siphoviridae*-like head-tailed viruses have already been reported for methanogenic digesters previously [29,33]. Moreover, a recent study on a *Siphoviridae* phage, designated Blf4 and isolated from a commercial biogas plant in Germany, also described that this phage infected methanogenic *Archaea* of the species *Methanoculleus bourgensis* [58]. Considering the relatively high abundance of *Methanoculleus* members in AD [19,59,60] and its metabolic relevance, Blf4 and *Siphoviridae*-like phages infecting members of the genus *Methanoculleus* genus might profoundly affect their host’s abundance and, thus, the efficiency of the entire biogas process.

Furthermore, among others, one *Tectivirus* (MAVG 515) was also identified within the metavirome. *Tectiviruses* are supposed to be one of the ancestors of the Polintons (large eukaryotic DNA transposons) that are suggested to be the first group of eukaryotic dsDNA viruses which have evolved from archaeal and bacterial ancestors [61,62].

Apart from the order *Caudovirales*, characterized by dsDNA genomes, the presence of viruses with ssDNA genomes in biogas-producing plants has not yet been demonstrated and has, therefore, been questioned. However, in this study, MAVG 157 of the family *Inoviridae* (order *Tubulavirales*), previously described as a virion with a circular, positive sense ssDNA genome [63], could be identified (Table 1). Members of this family have a unique morphology, visible as flexible filaments or rigid rods, caused by the helical symmetry of the capsid [64]. These phages infect Gram-positive, Gram-negative, or cell wall-less bacteria. A characteristic feature of this family is that its members neither enter typical lytic nor lysogenic cycles. Instead, virions are released from cells by extrusion, causing chronic infection without killing the host [64,65]. They mobilize DNA within the microbiome, and thus play a role in the evolution of microorganisms. MAVG 157 of the family *Inoviridae* is 8.471 bp long and showed threefold genome coverage, based on the sequencing data obtained. This is the first study so far indicating the occurrence of ssDNA phages in a biogas-producing microbiome.

### 3.4. Protein-Based Phage Similarity Network Showed Relationships with Phages Infecting Members of the Classes Bacilli and Clostridia

To predict the taxonomic relatedness of biogas plant phages with phages deposited in the ProkaryoticViralRefSeq v94 database and to indicate a narrow host range of virion studies, a genome- and network-based clustering classification method applying the program vConTACT v2.0 [49] was used (Figure 3). The resulting whole-genome clustering network was composed of 2991 phages in total. MAVGs were grouped based on their gene content profiles into viral clusters (VCs) related to candidate viral genera as assigned by the International Committee on the Taxonomy of Viruses [65]. The network revealed that biogas plant phages were spread across 65 VCs, with 75 additional phage genomes that were considered to be outliers, 134 singletons, and 13 MAVGs showing overlaps with other VCs. The latter are those whose genomes showed no common genomic features among all known viruses to justify the membership, and, therefore, are presumably specific to the biogas reactor environment. Moreover, these results suggest that the 357 MAVGs most likely represent novel viruses due to the low proportion of hits to reference phage genomes, and since they form VCs distinct from reference phages.

In total, nine MAVGs were affiliated with phages that infect bacterial host cells of the phyla *Firmicutes, Actinobacteria,* and *Proteobacteria*. Among the cluster related to *Proteobacteria,* only MAVG 101 (family *Podoviridae*) was related to phages that attack members of this phylum. A further three biogas plant phages were clustered with viruses infecting members of the phylum *Actinobacteria*. They originated from the vConTACT v2.0 database and belonged to the families *Podoviridae* and *Siphoviridae*.

Among the phylum *Firmicutes*, five biogas plant phages created viral clusters with other phages, infecting members of the classes *Bacilli* and *Clostridia*. These phages belonged to the family *Siphoviridae* or represented virions with unknown taxonomy. These results suggest that members of the classes *Bacilli* and *Clostridia* are most likely affected when phages initiate the lysis of host microorganisms, and thus might cause significant process disturbances due to the decimation of essential microbial groups within the AD.

### 3.5. Insights into the Biogas Plant Phages Life Cycle by Comprehensive Genome Analysis

To predict the lifestyle (temperate or virulent life cycle) of the phage community obtained from the mesophilic laboratory-scale biogas reactor, the 375 MAVGs of this study were analyzed using the Phage Classification Tool Set (PHACTS) [51]. Out of the 375 phages, PHACTS was able to confidently determine the lifestyle of 40 MAVGs confidently (Figure 4C). Unreliable results were obtained for the other 335 MAVGs; sometimes, these MAVGs were classified as virulent or as temperate at the same time. Most likely, the low proportion of the confidently predicted results was due to the novelty of biogas phage genome sequences in the database consulted for comparison. Furthermore, some of the obtained sequences were too short for meaningful comparative analysis, such as the protein-based phage similarity network analysis, as they might not play an important role for the phage lifecycle. In total, 14 out of 40 MAVGs were classified as virulent, whereas 26 MAVGs were predicted to follow a temperate life cycle (Figure 4C). Unfortunately, none of the confidently predicted phages represented the most abundant MAVGs of this study. Therefore, the ten most abundant and the two longest phages identified in this study were analyzed in detail to deduce their potential role within the studied biogas microbial community. Predicted open reading frames (ORFs) were initially classified according to nine different functional groups: DNA replication (e.g., DNA polymerase and exonuclease), host modification, putative functions, DNA modification (e.g., transposase), structural proteins (e.g., capsid proteins and tail proteins), host lysis, lysis-related proteins, auxiliary metabolic genes (AMGs), and packaging proteins (Figure 4A, Appendix A). A substantial fraction of the genes in MAVGs encodes hypothetical proteins (78.2%) with unknown functions, indicating the need for further experimental work addressing the functional characterization of phage-related genes. Genes encoding proteins involved in mandatory functions were also not further considered. On the other hand, genes involved in host modifications, host lysis, and lysogeny-related proteins were of particular interest and, therefore, were examined more closely. A gene involved in host modification was only found in MAVG 6, the largest phage genome in this study (Figure 4B). Its genome harbored a gene encoding an ADP–ribosyltransferase (ADPRT), which acts on the α subunit of the host RNA polymerase [66]. The activity of ADPRT was described to facilitate gaining control over the infected host cell primarily through shifting gene expression toward the phage genes at different stages of growth [66]. Brown and colleagues also assumed that many ADPRT-encoding genes are accessory genes that are horizontally transferred to confer a fitness advantage to the host [67,68]. Shmakov et al. also predicted that ADPRT is usually encoded by CRISPR-Cas systems loci, which might help to stabilize the prophages and plasmids in the host bacteria [69].

Furthermore, the phosphoadenosine phosphosulfate reductase (PAPS reductase) gene representing an AMG, was identified in 13 MAVGs studied (80, 112, 137, 189, 588, 62, 106, 149, 162, 303, 630, 631, and 655), four of them being among the ten most-abundant virions (MAVG 80, 112, 137, and 189), and MAVG 588 being the second-longest phage genome. AMGs are known to modulate the fitness of the phage by providing functions to the microbial organisms to improve the energy metabolism for phage reproduction [6,7]. AMGs featuring functional relevance in carbon metabolism, assembly of iron–sulfur clusters, nitrification, methane oxidation, and other metabolic processes are believed to give an advantage by improving utilization of nutrients within their host [8,9]. PAPS reductase has been described to be involved in assimilatory pathways of sulfate reduction, suggesting that phages can influence the metabolism and cycling of this essential element [70,71] and thus have an impact on the biogas microbial community (Appendix A). Furthermore, it was hypothesized that PAPS reductase is an accessory protein belonging to the clade of CysH family enzymes associated with type-IV CRISPR-Cas systems [69], representing the bacterial and archaeal adaptive immunity systems against virus particles [72]. Each of the PAPS reductases encoding MAVGs was identified as viral contigs, indicating that the identified PAPS gene is embedded in a phage region.

### 3.6. Occurrence of Phage Relatives in Biogas-Producing Microbial Communities as Deduced from Publicly Available Metagenome Data

To evaluate the prevalence or, rather, abundance of phages within the microbial communities of different biogas plants, 66 publicly available biogas metagenome datasets obtained from the NCBI database were mapped on the viral genome sequences using the program SParkHit. Fragment recruitments were performed by mapping a maximum of ten million sequences from the corresponding metagenome representing the biogas microbial community. The mapping approach was aimed at the reconstruction of phage genomes (represented by the MAVGs) from the previously published metagenome data. The metagenome fragment mapping results were distinguished into the following groups: (I) the occurrence of all 357 phage sequences in the metagenome and (II) the abundance of a single MAVG in the metagenome. In total, 27 out of the 66 metagenomes analyzed (≥0.1% of all metagenome sequences) were assigned to group (I) with different abundance values. Particularly high abundance values ranging between 0.5% and 1.8% of all metagenome sequences were detected for 7 out of the 66 metagenome data sets (accession numbers NCBI: SRR3184553 with (0.5%), ERR843252 (0.6%), ERR843254 with (0.7%), ERR843253 (0.8%), SRR2917898 (1.1%), SRR2917897 (1.3%), and SRR2917896 (1.8%)) (Figure 5).

The highest number of mapped sequences within group (I) was found in the metagenome originating from a production-scale biogas plant located near Cologne, Germany (between 1.1% and 1.8%; accession numbers SRR2917898 and SRR2917896, respectively). This biogas plant was operated under mesophilic (40 °C) conditions and fed with a mixture of maize silage (69%), cow manure (19%), and chicken manure (12%) [73]. Güllert and colleagues reported that a substantial number of cellulosome-producing bacteria of the phylum *Firmicutes* (57%), followed by *Bacteroidetes* (11%) and *Actinobacteria* (7%), were detected in the studied reactor [73]. Interestingly, the abundance values for phage sequences in this metagenome were unusually high (1.8%), because, on average, between 0.1% and 0.5% of viral sequences were previously detected in AD microbiomes [17]. Unfortunately, the analysis of the viral community was not an objective of the study by Güllert et al. [73]. However, our results indicated significant similarity between the phages presented in Güllert’s data and the metavirome of this study.

The biogas plant phage members were also detected with abundances between 0.5% and 0.8% in the microbiome originating from a mesophilic industrial biogas plant operating under dry fermentation conditions and fed with maize silage (63%), green rye (35%), and chicken manure (2%) (ERR843252, ERR843253, and ERR843254) [59]. As expected, *Clostridia* and *Bacilli* were also the most abundant classes of *Firmicutes* in the studied microbiome, with 14.3% and 1.4% of the total number of analyzed reads, respectively. As proposed in the previous chapter, phages from this study are most probably specific for *Firmicutes* members residing in mesophilic communities in particular.

Detailed analysis of fragment recruitment results also revealed that several of the analyzed MAVGs appeared to be highly abundant in mesophilic microbial communities, and were hardly found in microbiomes originating from thermophilic biogas reactor systems (group II). This trend, represented by the color change in Figure 5 (from blue to orange), indicates that the phages obtained in this study are most likely specific for mesophilic microorganisms residing in the biogas microbiome. MAVG 28 falls into this category and was found to be the most abundant virion among all MAVGs in this study. Its highest amount (0.1%) within the publicly available biogas metagenome datasets was detected in the mesophilic biogas plant studied by Stolze and colleagues [59]. Compared to the otherwise detected small proportion of viral sequences in biogas-producing communities being, on average, ≥1%, the abundance value achieved for MAVG 28 was surprisingly high. MAVG 28 was taxonomically classified as belonging to the family *Myoviridae*. Its genome is 44.297 bp long and was proposed to undergo the lytic cycle (non-confidently) of replication, as deduced from its genetic repertoire. MAVG 558 seemed to be less prominent in most of the analyzed microbiomes. However, this virion was the second most-abundant genome in the dataset described by Güllert and colleagues (between 0.044% and 0.074%) [73] and represented an unclassified phage with a genome length of 52,688 bp. Similar to MAVG 28, the MAVG 558 was proposed to encode the temperate cycle of replication (non-confidently).

The abundance values for the longest phage genome in the studied metavirome, MAVG 6, placed this virion on the fifth place in the ranking (0.040%, accession number SRR2917897). MAVG 6 was also found to be prevalent in the microbiome described by Güllert and colleagues [73], but appeared to be slightly abundant also in other mesophilic biogas-producing microbial communities (Appendix A). MAVG 6 was classified as belonging to the family *Siphoviridae*; however, the results obtained from the protein-based phage similarity network analysis indicated only a distant relationship with previously known phages. Together with 19 other MAVGs, MAVG 6 formed an independent cluster in the plot, which indicated the novelty of this virion (Figure 3).

To determine whether the biogas virome members were present in other habitats, metagenome fragment mappings were also carried out using publicly available metagenome data originating from cow dung, chicken gut, fish gut, pig gut, and soil environments. The phage abundance values achieved for these data were negligible, indicating a specific niche for the phages analyzed in this study, which is clear for the biogas plant environment.

Information obtained from the fragment recruitment approach showed that the MAVGs analyzed within this study can also be found in different biogas plants and most probably affect mesophilic microbial community members. Therefore, viruses may shape AD microbiomes.

## 4. Conclusions

Many of the environments in which phages interact with their bacterial hosts are anaerobic. Some of these scenarios are of importance regarding biotechnological applications, such as energy recovery by the anaerobic digestion of biomass. In the present study, high-throughput metavirome sequencing of samples from a laboratory-scale biogas reactor was performed. The aim was to provide an adapted protocol for the enrichment of bacteriophages from biogas digestates as preparation for phage–metavirome studies, and, in addition, to gain a better understanding of the phage’s community composition and its influence on the biogas microbiome. The protocol for the enrichment of virus-like particles from a biogas digestate was successfully established and serves as an orientation aid for other studies focusing on the isolation of bacteriophages from AD fermentation samples. Although this study does not fully reflect the complexity of the biogas-producing microbiomes, the obtained results provide some insights into the ecological role of phages within the analyzed biogas microbial community. Unfortunately, the interpretation of the generated data was limited due to the restricted availability of appropriate reference genome sequences in public databases, which prevented their complete characterization when relying on sequence homology-based identification. Accordingly, the enrichment and purification of phage particles applied in this study and subsequent sequencing led to the odentification of further key phage species, thus providing genome sequence information for novel phage members of the biogas-producing community.

The occurrence of phages in the biogas microbiome certainly has a significant impact on the composition of the biogas community and, therefore, the dynamics of biomass conversion. The obtained results indicated that *Firmicutes* followed by *Actinobacteria*, in particular, may severely be affected by phage infections. As bacteria from the phylum *Firmicutes* are predominantly involved in the hydrolysis of the fed substrate, it can be assumed that phages might strongly affect this step of the biomethanation process. Furthermore, several phages from this study are most probably specific for *Firmicutes* and *Actinobacteria* members residing in mesophilic communities in particular. Thus, further genetic analysis of the gene contents of the identified phage contigs could shed light on phage lifestyles in the biogas microbiome and expand the knowledge on the interactions between phages and their hosts under anaerobic conditions.

## Figures and Tables

**Figure 1 microorganisms-10-00368-f001:**
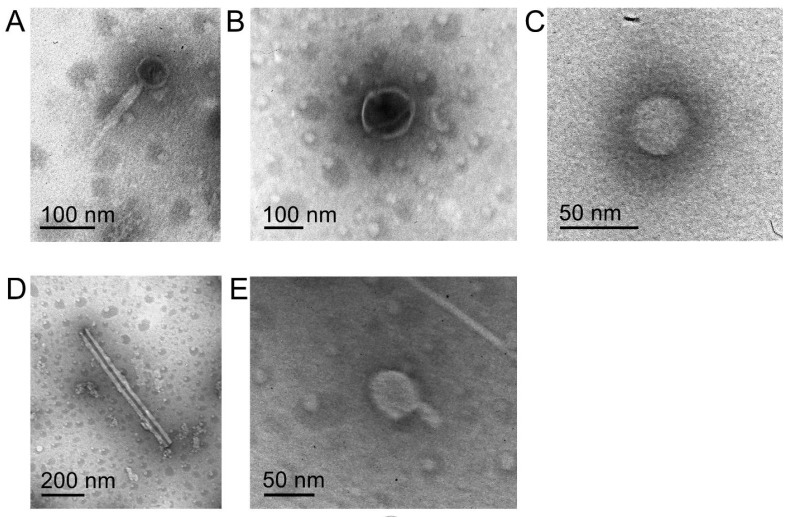
Morphological diversity of the phage-like particles observed in a sample originating from a laboratory-scale biogas reactor: (**A**) *Myoviridae*, (**B**) icosahedral phage-like particles, (**C**) *Podoviridae*, (**D**) *Rudiviridae*, and (**E**) *Myoviridae*.

**Figure 2 microorganisms-10-00368-f002:**
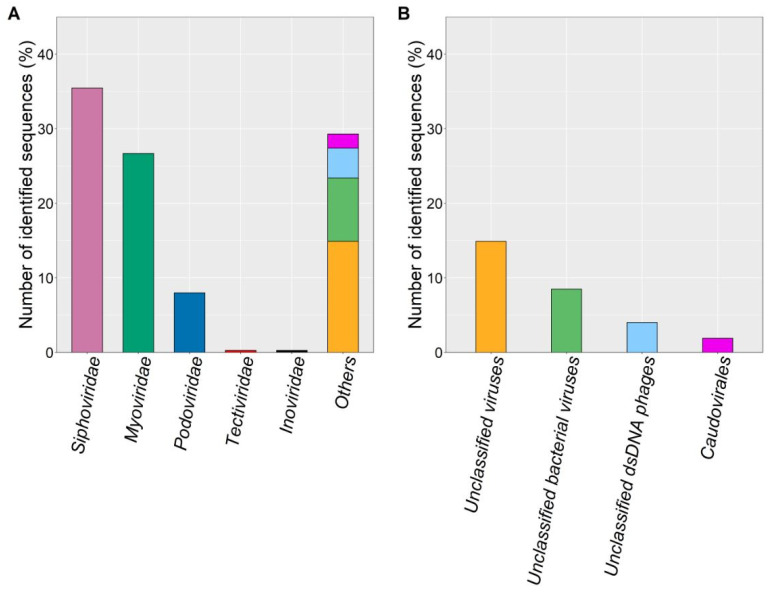
Taxonomic profiling of the biogas metavirome applying the “What the Phage” (WtP) pipeline [44]. The colors represent different families of phages as well as unclassified viruses. (**A**) Number of identified and classified sequences of metagenome-assembled viral genomes (MAVGs) that could be assigned to a certain family. (**B**) Number of identified and classified sequences and taxonomic classification of MAVGs assigned in Figure 2A as ‘others’.

**Figure 3 microorganisms-10-00368-f003:**
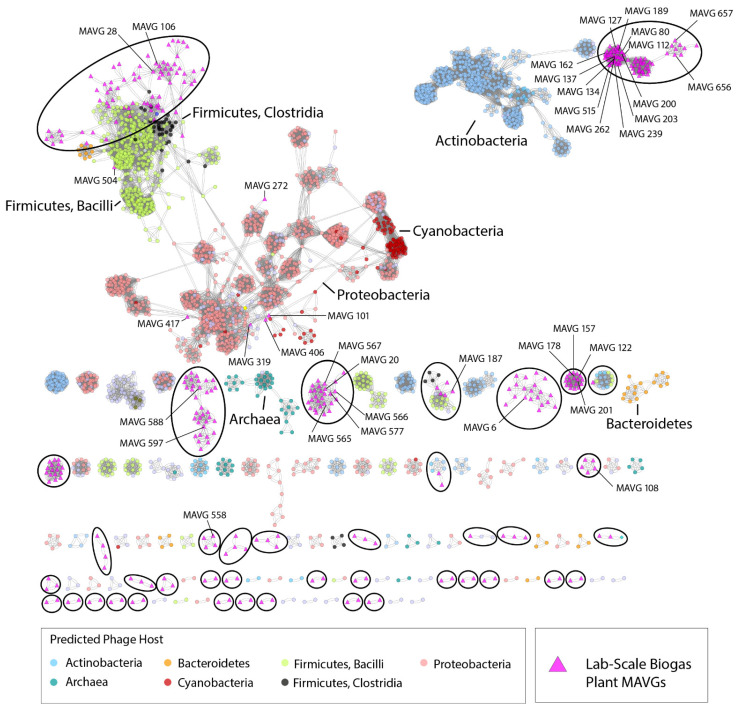
Protein-based phage similarity network constructed applying the vConTact 2.0 pipeline [49]. The vConTact 2.0 network generated by using the ProkaryoticViralRefSeq v94 virus database in conjunction with the phage dataset originating from a laboratory-scale biogas reactor (375 sequences) analyzed in this study using the perfused forced directed layout. Related bacteriophages are represented as triangles in pink with the edges between them representing shared protein clusters. Node colors were used to indicate phage hosts, whose taxonomic affiliations were positioned right next to the respective cluster. Ellipses around the groups indicate the biogas phages of this study. Network visualization was generated using Cytoscape 3.8.2.

**Figure 4 microorganisms-10-00368-f004:**
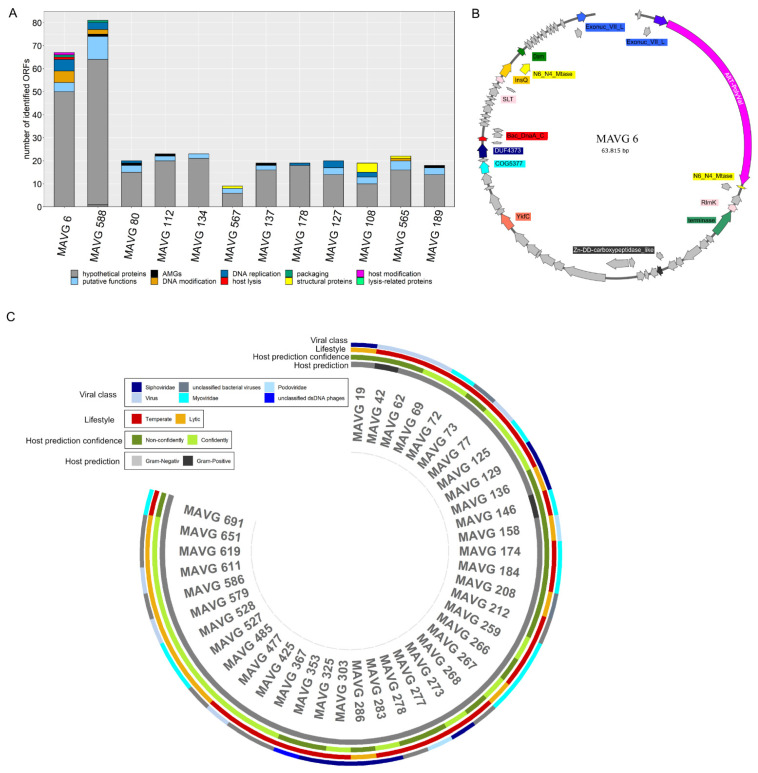
(**A**) Classification of the identified ORFs in selected MAVGs. ORFs were grouped into the categories: hypothetical proteins, AMGs, DNA replication, packaging, host modification, putative functions, DNA modification, host lysis, structural proteins, and lysis-related proteins. (**B**) Genomic map of MAVG 6 created with the Snapgene viewer tool. ORFs are represented by arrows and those encoding hypothetical proteins are shown in grey. The ADP–ribosyltransferase (ADPRT) gene, highlighted in pink, represents a genetic determinant involved in the modification of host proteins. (**C**) Representation of the viral class, lifestyle prediction, host prediction, and host prediction confidence generated by PHACTS for 40 MAVG genomes.

**Figure 5 microorganisms-10-00368-f005:**
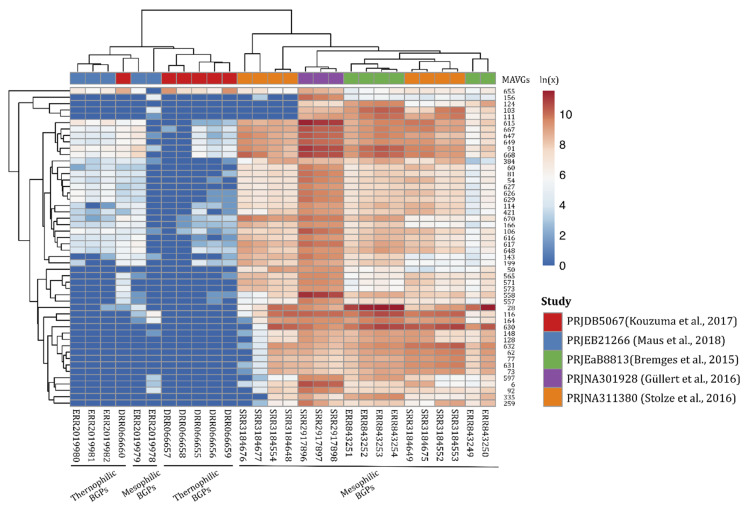
The 50-most-abundant MAVGs found in different microbial communities derived from publicly available metagenomes. Selected samples (the top) showing metagenome fragment mapping results of greater than 0.5% were visualized. Abbreviation: BGP, biogas plant.

**Table 1 microorganisms-10-00368-t001:** The most abundant phage genomes in the analyzed biogas reactor sample as deduced from metavirome sequencing.

Position According to the MAVG Abundance	Phage Contig ID	Contig Length [bp]	Coverage	Circular Genome	Number of Mapped Sequences ****	Taxonomy at the Family Level
1	MAVG 80	12,554	9083	no	16,283	*Siphoviridae*
2	MAVG 112	12,903	214	no	684	*Myoviridae*
3	MAVG 134	11,568	162	no	498	*Siphoviridae*
4	MAVG 567	10,561	156	no	428	*Myoviridae*
5	MAVG 137	10,247	149	no	389	*Siphoviridae*
6	MAVG 178	8780	90	no	214	unclassified
7	MAVG 127	11,531	84	no	285	*Podoviridae*
8	MAVG 108	12,998	79	no	271	*Siphoviridae*
9	MAVG 189	10,460	63	no	186	unclassified
10	MAVG 566	10,816	54	no	142	unclassified
11	MAVG 657	7709	52	no	136	unclassified
12	MAVG 200	7316	47	no	99	*Myoviridae*
13	MAVG 28	44,297	45	yes	560	*Myoviridae*
14	MAVG 187	8926	43	no	133	*Siphoviridae*
15	MAVG 577	9334	42	no	98	unclassified
16	MAVG 203	8666	42	no	98	*Siphoviridae*
17	MAVG 122	9252	42	no	112	*Siphoviridae*
18	MAVG 20	10,140	40	no	121	unclassified
19	MAVG 239	9882	40	no	117	unclassified
20	MAVG 21	8028	39	yes	84	*Siphoviridae*
21	MAVG 201	9048	37	no	104	*Siphoviridae*
22	MAVG 597	31,553	37	no	323	unclassified
23	MAVG 162	5791	35	no	96	*Siphoviridae*
24	MAVG 222	8975	34	no	90	*Myoviridae*
25	MAVG 593	36,077	34	no	405	*Myoviridae*
26	MAVG 202	8422	34	no	77	*Myoviridae*
27	MAVG 252	9143	34	no	94	*Siphoviridae*
28	MAVG 160	10,174	33	no	96	*Siphoviridae*
98	MAVG 6 *	63,815	13	yes	250	*Siphoviridae*
37	MAVG 588 *	52,688	29	yes	1	unclassified
51	MAVG 157 **	8517	24	yes	4	*Inoviridae*
94	MAVG 515 ***	5271	14	yes	2	*Tectiviridae*

* The two largest phages obtained in this study. ** Species harboring a ssDNA genome. *** Uniquely detected MAVG of the family *Tectiviridae* within the analyzed metavirome. **** The numbers of the mapped sequences represent the numbers of all sequences obtained after the ONT sequencing and polishing with the Illumina data using the Pilon program. The sequence information of all MAVGs of this study is deposited in the DDBJ/EMBL/GenBank database under the assembly accession numbers ERZ4966896.1-ERZ4966896.494.

## Data Availability

The metagenome data representing the biogas virome were deposited in the DDBJ/EMBL/GenBank database under the Bioproject ID PRJEB49144. Furthermore, the sequence information of all MAVGs of this study is deposited under the assembly accession numbers ERZ4966896.1-ERZ4966896.494.

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
