# Peer review of "Phage Genome Diversity in a Biogas-Producing Microbiome Analyzed by Illumina and Nanopore GridION Sequencing"

_microorganisms, 2022, doi:10.3390/microorganisms10020368_

Round 1

Reviewer 1 Report

The work of Willenbücher et. al. entitled "Phage genome diversity in a biogas-producing microbiome analyzed by Illumina and Nanopore GridION sequencing" describes matavirome with emphasis on phages in specific environments which are biogas-producing reactors. It is an interesting publication with a good experimental pipeline and data analysis. Some minor improvements are needed before publication.

I suggest publishing this work in Microorganisms journal with minor revision.

Comments and questions:

  1. Line 31 – “Tectavirus” – it is not the name of the phage family. It should be Tectiviridae.
  2. Lines 32-33 – “Using publicly available biogas metagenomes,…” should be changed to “Using publicly available biogas metagenomic data,…”
  3. Line 86 – describe the same as in previous examples:  e.g. “… particles e.g., Tobamovirus-like (family Virgaviridae)…”
  4. Line 91 – “in AD has” change to “in AD systems have”
  5. 2. Electron Microscopy section – in the Ackermann’s protocol phages are sedimented by centrifugation for 60 min at 25, 000 × g. However the authors decreased the speed to 18,000 x g. In my opinion such an approach could result in decrease of sedimented phages. Please discuss such differences in the preparation of phages for TEM analysis in the result section.
  6. Line 167 – change “rpm” to “g”.
  7. Lines 166-167 – I was confused reading this description. In the 2.2. section the authors centrifuged the sample for 60 min at 18, 000 × g for phage sedimentation (according to Ackermann’s protocol). However, in the 2.3 section the centrifugation of the sample “for 1 hour by 23.000 rpm (Beckman L-70 Ultracentrifuge)” did not result in phage sedimentation – the authors suggested that phages were present in supernatant. Please give an explanation of this discrepancy.
  8. Figure 1 and the description in the text (lines 250-253) – in my opinion the figure 1 presents: A – Myoviridae; B – I am not sure if this is a phage virion (in my opinion not); C – Podoviridae or head of Siphoviridae; D – Rudoviridae; E – Podoviridae; F -Myoviridae. It is difficult to classify phages based on only one picture of virion. The tail on the picture could be “hidden” or broken, and only part of the tail could be visible. I suggest rewriting the text.
  9. Why on figure 1 can we not see phages from the Siphoviridae family? In metagenomic analysis such phages represent the most abundant group of phages in the reactor (e.g. MAVG 80). It must be discussed and explained.
  10. Lines 279-280 – “Details of the MAVGs genomes are listed in the supplementary information (Supplementary Material Table S1)”. The most important information is not included in Table S1. The authors have to show the obtained sequences of phage genomes/fragments. Such data should be deposited in public databases.
  11. Figure 2 – The relative abundance should be present as a number of identified sequences. This is the result of obtained and sequenced number of phage sequences. We do not know if 100% of phages present in the sample were sequenced. It will be clearer for the readers.
  12. Line 312 and 313 – Tectavirus? Tectaviruses? It should be Tectivirus and Tectiviruses (from the Tectiviridae family as present on Fig. 2).
  13. Table 1 – add accession numbers to the phage sequences and include into the manuscript the accession no. to all obtained sequences.
  14. Lines 351-352 – I do not see any phages infecting Actinobacteria (on Fig. 3). If there are some, please indicate them on Fig. 3.
  15. Caption of the figure 3 – there are numbers of colors on the figure representing various phage hosts. However, there is no legend or description for all of these colors. It makes the figure unclear for the readers. Please add clear information which color represents which bacterial host.
  16. Lines 377-379 – “Most likely, the low proportion of the confidently predicted results is due to the novelty of biogas phage genome sequences in the database entry to be compared.” In my opinion there is another reason – some of the obtained sequences are too short for such analysis and/or represent genome regions which do not play a role in the phage lifecycle. It could be included in the discussion in this section.
  17. 5 – what is presented at heatmap on the right (range from 0 to 10)? It should be clearly described.
  18. Line 449 – a spell mistake in acc. No. There are two the same numbers (SRP2917896).
  19. Lines 490-494 – The names of MAVGs in the table S4 are different. If I understand the description correctly the MAVG 6 is named as “tig00000006”. The names should be the same as in the manuscript text. Additionally, there is no information about the numbers in the table S4 – it should be clearly described what the numbers present. Moreover, MAVG 6 has the highest number (is it % or other units?) in metavirome SRR2917896 (not in SRR2917897).
  20. Data Availability Statement – I can not find any data under the Bioproject ID PRJEB49144. Such data have to be published with the manuscript for the reviewer and other future readers.

Author Response

Dear reviewer,

We thank you for your positive evaluation of our manuscript and your constructive criticism. Please find attached our point-to-point response to your comments.  

Reviewer 2 Report

  1. The intended meaning of the following line in the abstract is unclear. Are bacteriophages not classified in the super kingdom virus ?

‘ Two-thirds of the classified sequences were assigned to the superkingdom Viruses, with the remaining one-third belonging to the family Siphoviridae, followed by Myoviridae, Podoviridae, Tectavirus and Inoviridae’

  1. Change the word parasitize to infect in the abstract in the following line ‘phage genomes that parasitize members of the classes Clostridia and Bacilli’
  2. Line 59-60: There is a published study of the termite gut metavirome which is represents an anaerobic biomass degrading environment. This should be cited.
  3. In material and methods, The phage precipitation part is not clear. The first centrifugation was at same rpm for 1 hour and the second one was also at same rpm for four hours ? Is there a possibility that some phages may have been lost in the first centrifugation ? Was there PEG/NACL added to precipitate the phages? Is there any reference where a similar method was used ?
  4. The title focuses on illumina and nanopore sequencing, but in the manuscript the purpose of using two different sequencing technologies is not described clearly.
  5. In table 1, do the number of mapped sequences represent read numbers or assembled contigs ?
  6. How are the MAVGs classified in table 1? Does what the phage pipeline provide classification ?
  7. How many MAVGs were identified ? 357 or 375 ? Line 283 and 285 show different numbers. The supplementary table has 380 MAVGs.
  8. I highly recommend authors to add a phylogenetic tree of the all the terminase genes (amino acids). This will give a good picture of the entire diversity of phages in the sample in one image.
  9. There is no information about the 103 genomes that could not be classified further than the super kingdom. However, in the supplementary table 1, 34 are classified as virus or viruses. 10 are unclassified. 32 are unclassified bacterial viruses. 15 are environmental samples and 15 are unclassified dsDNA phages. Authors should clarify this. 

Author Response

(The authors gave the same response as above.)

Reviewer 3 Report

Response to Research article

Phage genome diversity in a biogas-producing microbiome analyzed by Illumina and Nanopore GridION sequencing

The author presents a good analysis of phage genome diversity in a mesophilic anaerobic digester. Briefly, the diversity of the phage community based on transmission electron microscopy (TEM), genomic analysis was studied along with identification of their lifecycle. Based on the protein-based network analysis, the host range of the phage was deduced as well. Additionally, metagenome data from other biogas plants were used to examine the relatedness of extracted phage sequences. The result highlights that the phage sequences extracted are specific to mesophilic conditions rather than thermophilic conditions or other environments (dung, gut, soil ecosystem).

This study uses the combination of MinION and Illumina sequencing, which is different from other studies. However, some of the details of the analysis are lacking. Especially the procedure highlighting the polishing of contigs is missing. Also, it is not clear how the MAVGs were extracted. 

Some of the detailed comments are mentioned below:

  • L126-149: How are these operational conditions related to the genomic analysis?
  • Based on the results presented, it looks like only one sample was taken. However, in the methods section, “samples” have been mentioned multiple times. I think it should be made clear.
  • L229: How exactly was the MAVG extracted? Is it just the assembled contigs or a group of contigs? The detailed procedure is not clear in the methods section.
  • L268: 32.3 instead of 32,3
  • L269: What was the expected diversity? Any reference? 
  • L272: What exactly was performed to polish the contigs using the Illumina data?
  • L281-329: The identification of the phage sequences was related to the potential hosts. Since phages are host-specific, generalizing the phage-host association based on the family level is tricky. For example, the phage of the family Siphoviridae (L300-311) has been speculated to infect Blf4 based on other studies. This is an extrapolation of results. It would have been better if the phage host association was verified with downloaded sequences of the host and then connected with the phage sequences using different published bioinformatics algorithms.  
  • L371-L421: The AMGs identified in the phage genome don’t necessarily mean it is aiding towards the host’s fitness if the host doesn’t require the gene. Therefore, it would be important to identify the strain of the host that the phage is infecting.
  • Figure 5: It is not clear in the standalone Figure 5 what the color gradient in the heatmap represents. Please elaborate on that in the figure or the captions.

Minor comments

  • L53-55: The two sentences should be merged together.
  • L167, L169: 23,000 rpm instead of 23.000 rpm

Author Response

(The authors gave the same response as above.)
